# Assessment of Nutritional Intakes in Individuals with Obesity under Medical Supervision. A Cross-Sectional Study

**DOI:** 10.3390/ijerph16173036

**Published:** 2019-08-22

**Authors:** Costela Lăcrimioara Șerban, Alexandra Sima, Corina Marcela Hogea, Adela Chiriță-Emandi, Iulia Teodora Perva, Adrian Vlad, Alin Albai, Georgiana Nicolae, Salomeia Putnoky, Romulus Timar, Mihai Dinu Niculescu, Maria Puiu

**Affiliations:** 1Functional Sciences Department, “Victor Babes” University of Medicine and Pharmacy Timișoara, 14 Spl. Tudor Vladimirescu, 300172 Timişoara, Romania; 2Department of Internal Medicine—Diabetes, Nutrition and Metabolic Diseases, “Victor Babeș” University of Medicine and Pharmacy, 156 Liviu Rebreanu Blvd., 300723 Timişoara, Romania; 3Center and Clinic for Diabetes, Nutrition and Metabolic Diseases, “Pius Brinzeu” Emergency County Hospital Timisoara, 156 Liviu Rebreanu Blvd., 300723 Timişoara, Romania; 4Genetics Discipline, Center of Genomic Medicine Timișoara, “Victor Babeș” University of Medicine and Pharmacy, 2 Eftimie Murgu Square, 300041 Timișoara, Romania; 5“Louis Țurcanu” Clinical Emergency Hospital for Children, 2 Iosif Nemoianu St., 300011 Timișoara, Romania; 6NutritioApp—Advanced Software for Nutrition, 19 Pridvorului St., bl. 20, sc. 1, ap. 2, 300172 Bucharest, Romania; 7Pediatric Cardiology Department, Emergency Children’s Hospital “Marie Curie”, 20 C-tin Brancoveanu Bd, 041451 Bucharest, Romania; 8Microbiology Department,” Victor Babes” University of Medicine and Pharmacy Timișoara, 16 Victor Babeș Blvd, 300226 Timișoara, Romania

**Keywords:** adequate intake, dietary reference intake, obese adult, nutritional epidemiology

## Abstract

People with obesity in Romania are often under medical supervision, which is aimed to decrease body weight and treat accompanying metabolic disorders and cardiovascular implications. However, there is limited information regarding the implementation of dietary recommendations in adults with obesity. We aimed to evaluate the prevalence of reaching the recommended intakes of macro- and micro-nutrients in adults with obesity under medical supervision. Individuals with obesity, recruited in the context of a study with a larger scope (NutriGen ClinicalTrials.gov NCT02837367), who were under medical supervision underwent four 24 h recalls in order to assess daily food intakes. Macro- and micro-nutrient intakes were computed, and the prevalence of reaching recommended dietary allowances (RDAs) for each nutrient was calculated. The majority of subjects did not meet the recommended intakes for most nutrients. Energy from fat exceeded the threshold of 35% recommended intake, even in the lowest quartile of energy intake. The micronutrients with less than 5% of individuals reaching the RDAs were vitamin D, vitamin E, fluoride, and omega-3 fatty acids for both males and females, and choline, magnesium, and potassium in females. The burden of inadequate nutrition in individuals with obesity should be acknowledged and properly addressed within efforts to reduce obesity rates and associated disorders.

## 1. Introduction

The increasing burden of obesity worldwide, along with its array of associated diseases, is one of the greatest global health challenges of our time. Consequently, considerable resources are used in trying to reduce obesity rates and mitigating its consequences upon health. However, overweight people and obesity are still increasing, and so is the associated morbidity, despite these efforts [1,2,3]. Nutritional management is the most employed type of intervention involved in weight reduction for people with obesity, alone or in combination with other interventions such as exercise and motivational counseling or other behavioral interventions [4]. While the role of macronutrients is well established and described in obesity-related nutritional management, there are limited and contradicting data regarding the management of micronutrient requirements during these interventions in both children and adults [5,6]. Although the importance of micronutrient adequacy is well recognized in obesity prevention and management [7], there is also limited information regarding the management of micronutrients in healthcare systems from developing countries.

In Romania, the prevalence of obesity, reported by recent published studies, varies largely between 9.1% and 34.70% [8,9,10], however these data are less reliable, as previously discussed [9].

Equally important, the Romanian healthcare system does not have a clear policy regarding the involvement of licensed nutritionists/dieticians in addition to the medical services offered to patients with obesity, and in many cases the health care system does not cover the expenses encumbered for employing dieticians along with the doctors in designing life-style interventions that are aimed to decrease body weight. As a consequence, patients with obesity, especially in outpatient settings, often do not receive structured advice from their health care providers about how to implement healthy dietary habits, or no dietary advice is offered at all. Hospitalized patients may receive adequate nutritional advice from hospital-associated dieticians, however there is no clear policy regarding follow-up once a patient leaves the hospital. In all such cases, the burden of designing and implementing healthy dietary habits remains almost completely on the patient’s shoulders, as the choice of further involving a registered dietician is made by a minority of patients who can afford the associated expenses.

This study aimed to characterize the adequacy of macro- and micro-nutrients consumed by adults with obesity that were, at the time of assessment, under medical supervision by a qualified healthcare provider for obesity-associated diseases. The tested hypothesis was that, in the absence of proper nutritional assessment for intakes and of structured dietary advice and follow-up throughout the period when subjects attempt to lose weight, it is unlikely that adequate nutritional intakes would be reached at the levels of recommended dietary allowances (RDAs), per either United States Department of Agriculture (USDA) [11] or European Food Safety Authority (EFSA) [12] criteria.

## 2. Methods

### 2.1. Recruitment of Subjects

421 adults with obesity (204 men and 217 women), who were at the time of recruitment under medical supervision due to obesity or various complications of obesity, were recruited in the context of a study with a larger scope (NutriGen, ClinicalTrials.gov NCT02837367), undertaken in Timișoara, Romania. Inclusion and exclusion criteria are presented in Table 1. The Scientific Research Ethics Committee Board of The Victor Babes University of Medicine and Pharmacy Timisoara approved this study (approval number: 6/20.06.2016), which was conducted in accordance with the Helsinki Declaration. Informed consent was obtained from all study participants.

### 2.2. AUDIT-C Assessment for Alcohol Abuse

In order to exclude alcohol abusers, all prospective subjects completed a questionnaire regarding their habits of alcohol consumption. According to the current guidelines for a healthy life style, the maximum recommended alcohol intake is one drink (10 g alcohol) per day for women and two drinks (20 g alcohol) per day for men [13]. This evaluation used a questionnaire (translated into Romanian language) adapted from the Alcohol Use Disorders Identification Test (AUDIT), designed by the World Health Organization. This contains 10 questions aimed to screen for hazardous and harmful alcohol consumption [14]. All prospective participants who had a score above 8 were excluded from participating in the study. The threshold of 8 was used because of its previous validation for being an indicator of hazardous or harmful drinking [15,16].

### 2.3. Dietary Assessment

Daily nutrient intakes were estimated using a 5-pass 24 h dietary recall, administered by trained medical doctors or volunteer medical students. The questionnaire was adapted in Romanian after a 24 h dietary recall, as previously published [17]. The 5-pass approach was validated elsewhere [18], and consisted of: (1) building a list of foods and beverages consumed in the past 24 h; (2) verification of the list; (3) obtaining further details about each item that was collected; (4) commonly forgotten foods were queried (such as sauces or other added ingredients, and water consumed between meals); and (5) the data were reviewed with the participant again for a final check of completeness. The recall was administered four times to each participant in four non-consecutive days, including a weekend day and 3 work days, by face-to-face interview (first interview) or by telephone (recalls 2–4). In order to determine the consistency of dietary intakes obtained from the administration of 24 h dietary recalls in 4 different sessions, the data retrieved for all nutrients (see below) were subject to paired t-test analysis, with appropriate false discovery rate (FDR) adjustments between any combination of two recalls. The dietary assessment evaluated what and how much of each food was eaten on the previous day, using common measures, in order to estimate portion sizes as accurately as possible. To minimize interviewer’s bias, all personnel involved (interviewers) followed a training course prior to the administration of the dietary recall.

### 2.4. Coding of Nutrient Intakes

For each day investigated, all foods and drinks, in amounts declared by participants, were converted to energy, macro-, and micro-nutrients using a web-application (Nutritio, Naturalpixel SRL, Bucharest, Romania, https://nutritioapp.com) [19]. The developers designed specific algorithms for batch data retrieval using the best match or decomposing some complex foods into composite dishes based on the USDA Food and Nutrient Database for Dietary Studies and for local foods. In order to avoid the bias introduced by the mandatory folic acid fortification in the US, the foods fortified with folic acid were replaced by identical or similar non-fortified items from other European databases.

### 2.5. Data Analysis

From the total of 421 adults recruited, 12 participants either withdrew from the study or were excluded due to incomplete data, leaving 409 subjects (197 men and 212 women) in the analysis. The status of 24 h dietary recalls for all participants is indicated in Table 2.

Data were analyzed using IBM-SPSS version 18 (IBM, Armonk, New York, USA). For all subjects included in the analysis, the average value of the 24 h recalls collected was calculated for all macro- and micro-nutrients exported from NutritioApp, which was further used in all analyses, since the within comparisons between any two recalls, adjusted with the Sidak method, were not significantly different (data not shown).

Energy intake was categorized into quartiles based on the data distribution, and within each quartile the percentages of energy from fat and carbohydrates were calculated.

Where Dietary Recommended Allowance (RDAs) were available, intakes of macro- and micro-nutrients were compared against these values, recommended by the European Food Safety Authority (EFSA, EU) and the United States Department of Agriculture (USDA, USA), and the percentages of individuals reaching these requirements for each nutrient were calculated.

For numerical variables, descriptive summary measures of central tendency (mean) and of dispersion (standard deviation) were computed. For categorical variables, frequency (%/N) was computed. Since all mean intakes failed for the assumption of normal distribution and homogeneity of variance, for comparisons between genders, the Man-Whitney test was used. For trend evaluation between quartiles, the Kruskal Wallis test was used for proportions and 1-way ANOVA for means. Chi-square test was used for comparing proportions between males and females for adequacy of nutrient consumption according to EFSA and USDA criteria.

## 3. Results

### 3.1. Mean Micronutrient and Energy Intakes

Mean daily intakes of micronutrients, fatty acid categories, and water, stratified by gender, are presented in Table 3. When comparing male and female intakes, except for fluoride, which was not significantly different between genders, all other comparisons indicate higher intakes in males. Percent energy from polyunsaturated fatty acids (PUFA) and saturated fatty acids (SFA) were not significantly different between genders.

Analysis of energy intakes indicated that when classified by quartiles, the increase in calories was associated with an increase of percent energy from fat and a decrease of percent energy from carbohydrates in both males and females (Table 4). For males, the percent of energy intake from fat ranged from 35.00% ± 7.84 (average for quartile 1) to 39.3% ± 6.89 (average for quartile 4) and the trend across quartiles was positive and close to statistical significance (*p* = 0.051). Conversely, the trend for percent energy from carbohydrates was negative (*p* = 0.010). In females, the percent energy intake from fat ranged from 33.1% ± 7.24 (average for quartile 1) to 40.7% ± 7.91 (average for quartile 4). An opposite trend was observed for the percent of energy derived from carbohydrates, which decreased with increased calorie intake.

### 3.2. Adequacy of Nutrient Intakes

The estimated intakes were assessed for adequacy using both the RDAs set by USDA and the dietary reference values (DRVs) set by EFSA. For the percent of energy derived from fat and carbohydrates, the RDA and DRV recommendations are identical. The fat and carbohydrate intakes were classified into three groups (low, recommended, and high) according to the RDA intervals for each group of nutrients. The statistical analysis tested whether these intakes were different between genders across quartiles of energy intakes (Table 5). Except for males on both energy derived from carbohydrates and energy derived from fat, in women, comparisons for trend were significant, indicating that the higher the energy intakes, the more individuals exceeded the RDA for fat, while the percent of individuals below RDA for carbohydrates increased. There were no differences between males and females when compared within each quartile of energy intake. For all participants, the percent of subjects with intakes within the RDA intervals ranged from 33.0% (males having fat intakes within RDA for energy derived from fat) to 53.3% (females having carbohydrate intakes within the RDA for energy derived from carbohydrates).

Some micronutrients, fiber, protein, and water intakes differ in their recommended intakes between USDA and EFSA. Therefore, we assessed the adequacy of these nutrients based on RDAs from USDA and DRVs from EFSA, separately. Table 6 indicates, for each micronutrient, fiber, protein, and water intake, the percent of participants, stratified by gender, who reached the recommended intakes for each nutrient. According to the USDA recommendations, the percent of individuals reaching RDAs ranged between 0% (both males and females for fluoride) and 86.80% of males reaching the RDA for iron and selenium. According to EFSA recommendations, the adequacy ranged between 0% (both males and females for fluoride) and 99.06% of females reaching the DRV for thiamine. Significant differences were identified between genders according to at least one criteria (USDA or EFSA) for the percent of individuals reaching the recommended intakes for the following nutrients: vitamin K, thiamine, riboflavin, niacin, vitamin B6, folates, vitamin B12, pantothenic acid, choline, calcium, copper, iron, magnesium, manganese, phosphorus, selenium, potassium, sodium (bellow upper limit), fiber, protein, and water.

## 4. Discussion

This study examined the daily dietary intakes for more than 30 nutrients in both male and female adults from Western Romania with obesity, who at the time of the assessment were under medical supervision, all having metabolic disorders or cardiovascular diseases. The study further assessed the prevalence of reaching adequate intakes for micronutrients and macronutrient adequacy and distribution as a source of energy, using both USDA and EFSA criteria.

One of the main findings was that the intake of micronutrients, according to both criteria, was below recommendations for the vast majority of obese adults. Because the participants were, at the time of our investigation, under medical supervision, we decided not to exclude any subject due to low/high estimated energy intake (mean energy intake in the lowest 5% range was 653.8 ± 105.4 kcal and in the highest 5% range was 3371.2 ± 840.9 kcal). It is important to note that for all participants, there was no long-term structured dietary advice given by licensed dieticians/nutritionists, nor follow-up plans intended to re-assess recommendations, and that the food intakes were, in general, a result of advice given by doctors, without specifics, or a result of voluntary decisions taken by the participants in an attempt to lower their body weight. It is worth mentioning that in Romania, the profession of a nutritionist or dietician has been acknowledged since 2015 [20], however Methodological Norms for the Application of the Law have been published only very recently, in 2019 [21].

Using the USDA criteria, the adequacy of intake for less than 5% of participants was observed for vitamin D, fluoride, choline, vitamin E, potassium, linoleic acid, and alfa linolenic acid in both genders, and for magnesium in females. When EFSA criteria were applied, the adequacy of intake for less than 5% of participants was noted for vitamin D, vitamin E, and fluoride in both genders, for alpha-linolenic acid in males, and for choline, potassium, and fiber in females.

Conversely, the highest percentages of subjects reaching the recommended intakes were for thiamine (ranging from 45.75% to 99.06% across gender or criteria stratification), iron (from 26.42% to 86.80%), and selenium (ranging from 35.38% to 86.80% across gender of criteria stratification). It is also important to note that for most nutrients, there were significant differences in the prevalence of reaching the recommended intakes between males and females (Table 6).

The estimated micronutrient intakes, as well as the prevalence of reaching the recommended values, are in line with previously published studies and point to the fact that obesity is associated not only with well-known associated diseases, but also with important and numerous micronutrient deficiencies, and in different settings (whether under supervised dietary interventions or not) [4,22,23,24,25]. However, this is the first study, to the best of our knowledge, that assessed micronutrient intakes in people with obesity who were under medical supervision and that, at best, may have received recommendations mainly focused on the macronutrient content of food, but with no implemented follow-up plans.

This study also indicated a misbalance in macronutrient energy sources, with varying intakes of energy from fat and carbohydrates (Table 4 and Table 5). While the recommended intake range of energy from fat was reached by 61.8% of subjects in the first quartile for energy intakes, only 23.5% of participants had an adequate value of energy from fat in the highest quartile, with no differences between men and women. These data also suggested that higher energy intakes were associated with increased consumption of fats.

The analysis of dietary fatty acid types revealed that 7.1% ± 2.4 from energy came from PUFAs and 12.7% ± 3.9% came from SFAs. These results are consistent with previous reports on fat and fatty acid intakes in European countries and worldwide. These indicated that in most countries, SFA intakes were higher than recommended, with intakes ranging from 8.9% to 15.5% E for SFA, while PUFA intakes were often below the optimal intakes, ranging from 3.9% to 11.3% E [26,27,28]. In an epidemiological review (26) in the general population, Romania was placed in the top 3 nations for SFAs intake. Globally, the estimated worldwide average intake of SFAs was 9.4% of energy intake (% E), with a 95% confidence interval between 9.2% E and 9.5% E. The same review found that globally, omega-6 polyunsaturated fat mean intake was 5.9% E, and mean intake of seafood omega-3 fats was 163 mg/day, while plant omega-3 consumption was 1371 mg/day.

Health problems arise not only from the low intakes of PUFA, but also from an inadequate ratio of LA/ALA. High omega-6/omega-3 ratios were previously associated with weight gain and mood and cognition disorders such as neurological and psychiatric disorders, including depression, anxiety, schizophrenia, and attention deficit hyperactivity disorder, in animal and human studies [29,30].

Sodium intake below the upper limit set by USDA was reached by only 10.6% of males and by 40.1% of females, and when using EFSA criteria, only 2.1% of males and 13.21% of females had lower intakes than the upper limit. Taking into account that when using sodium biomarkers usually an underestimation is detected, these results might be underestimating the real intake for this population [31]. In a previous report, the average sodium intake in Eastern Europe was 11.1 g salt (NaCl) per day [32].

This study has several limitations that need consideration. The dietary intakes were self-reported and underreporting of intakes (recall bias) is a common limitation of 24 h dietary recalls [33]. Since the study was cross-sectional, it did not infer causal relationships with current diagnostics.

Another limitation is that the accuracy of nutrient intake estimates was limited by the variability in both natural and processed foods as well as in laboratory analyses of food samples, which are in the databases used by Nutritio for nutrient retrieval. This study did not use biomarkers and therefore no methodological validation was possible. The data presented cannot be considered representative for all individuals with obesity, but could be, at best, representative only for those who decided to lose weight in the absence of professional, long-term dietary advice and supervision. Therefore, for people with obesity who are either under professional nutritional advice or who do not limit their calorie intake in order to lose weight, one cannot make inferences regarding the adequacy of micronutrient intakes.

## 5. Conclusions

In conclusion, this is the first study, to our knowledge, that assessed micronutrient and fatty acid intakes in a Romanian population, specifically in people with obesity under an attempted hypocaloric diet. When individuals with obesity were under medical supervision for obese-related diseases, but without dietary specific advice or being supervised by a trained dietician, the voluntary calorie reduction was associated with numerous micronutrient deficits and imbalanced fatty acid intakes. These deficiencies could add to the health burden in these individuals. Long-term dietary advice and supervision should be considered as an intrinsic part of the health care system involved in the reduction of the obesity-associated public health burden.

## Figures and Tables

**Table 1 ijerph-16-03036-t001:** Recruitment criteria.

Inclusion Criteria	Exclusion Criteria
Age 18–70 years. Obesity (BMI ≥30 kg/m^2^). Abdominal circumference ≥84 cm in women and ≥90 cm in men. Dyslipidemia: Total serum cholesterol ≥200 mg/dL. HDLc ≤50 mg/dL in women and ≤40 mg/dL in men. Serum triglycerides ≥150 mg/dL. Treatment for dyslipidemia (e.g., statins, fibrates, omega 3 fatty acids, cholestyramine, ezetimibe) or for Type 2 diabetes.	Diagnostic of any type of cancer, or medical history of cancer. Any auto-immune disease. Any diagnosed psychiatric disorder (except for situational/transient depression). Blood coagulation disorders, diagnosed in the present or in the past. History of drug abuse. Alcohol abuse evaluated using AUDIT-C.

HDLc, high-density lipoprotein cholesterol; BMI, body mass index.

**Table 2 ijerph-16-03036-t002:** Status of 24 h dietary recalls.

Number of Recalls	Patients with Recalls Numbers
4 × 24 h recalls	388 × 4 = 1552
3 × 24 h recalls	5 × 3 = 15
2 × 24 h recalls	4 × 2 = 8
1 × 24 h recalls	12 × 1 = 12
	409 participants imputed in Nutritio app = 1587 recalls

**Table 3 ijerph-16-03036-t003:** Mean daily dietary intakes.

Nutrient	Male	Female	*p* Value *	All Participants
Vitamin C (mg)	68.3 ± 51.1	55.0 ± 38.7	**0.029**	61.4 ± 45.5
Vitamin D (IU)	86.7 ± 80.5	57.6 ± 70.1	**<0.001**	71.6 ± 76.6
Vitamin A (µg)	649.3 ± 678.8	491.8 ± 439.4	**0.024**	567.7 ± 572.2
Vitamin K (µg)	92.0 ± 91.2	66.0 ± 68.7	**<0.001**	78.6 ± 81.3
Thiamine (mg)	1.6 ± 0.7	1.2 ± 0.5	**<0.001**	1.4 ± 0.6
Riboflavin (mg)	1.4 ± 0.6	1.1 ± 0.5	**<0.001**	1.2 ± 0.6
Niacin (mg)	19.9 ± 8.8	13.8 ± 6.6	**<0.001**	16.7 ± 8.3
Vitamin B6 (mg)	1.6 ± 0.7	1.1 ± 0.5	**<0.001**	1.23 ± 0.6
Folates (µg)	312.0 ± 149.2	225.8 ± 106.6	**<0.001**	267.3 ± 135.8
Vitamin B12 (µg)	4.6 ± 3.8	3.0 ± 2.9	**<0.001**	3.8 ± 3.5
Vitamin E (mg)	5.1 ± 3.0	4.1 ± 2.2	**0.001**	4.6 ± 2.7
Pantothenic acid (mg)	4.3 ± 1.7	3.4 ± 1.4	**<0.001**	3.9 ± 1.6
Choline (mg)	282.8 ± 133.0	207.0 ± 91.1	**<0.001**	243.5 ± 119.3
Iron (mg)	13.4 ± 5.6	10.0 ± 4.2	**<0.001**	11.6 ± 5.2
Calcium (mg)	1105.9 ± 517.2	881.1 ± 451.9	**<0.001**	989.4 ± 496.8
Magnesium (mg)	260.5 ± 115.5	193.6 ± 80.6	**<0.001**	225.8 ± 104.3
Fiber (g)	18.2 ± 8.5	14.3 ± 5.6	**<0.001**	16.2 ± 7.4
Copper (mg)	1.1 ± 0.5	0.8 ± 0.3	**<0.001**	0.9 ± 0.4
Fluoride (µg)	429.7 ± 448.3	349.1 ± 376.2	0.061	388.0 ± 414.0
Phosphor (mg)	1050.9 ± 426.4	773.6 ± 302.7	**<0.001**	907.2 ± 392.4
Manganese (mg)	2.2 ± 1.8	2.1 ± 6.6	**<0.001**	2.1 ± 4.9
Selenium (µg)	94.4 ± 36.6	64.8 ± 29.6	**<0.001**	79.0 ± 36.3
Zinc (mg)	8.0 ± 3.3	5.6 ± 2.5	**<0.001**	6.8 ± 3.1
Potassium (mg)	2424.4 ± 831.7	1951.6 ± 550.9	**<0.001**	2179.3 ± 738.4
Sodium (mg)	3852.5 ± 1684.6	2648.0 ± 1075.6	**<0.001**	3228.2 ± 1524.7
EPA (mg)	44.8 ± 100.7	26.1 ± 66.3	**0.001**	35.2 ± 85.1
DHA (mg)	91.3 ± 17.5	46.5 ± 92.5	**<0.001**	68.1 ± 140.0
LA (mg)	3.8 ± 3.8	2.2 ± 2.2	**<0.001**	3.0 ± 3.2
ALA (mg)	0.4 ± 0.5	0.2 ± 0.3	**<0.001**	0.3 ± 0.4
Fatty acids total saturated (g)	27.0 ± 13.9	18.8 ± 11.2	**<0.001**	22.8 ± 13.2
% energy from PUFA	7.1 ± 2.6	7.1 ± 2.2	0.459	7.1 ± 2.4
% energy from SFA	13.0 ± 3.9	12.4 ± 3.9	0.129	12.7 ± 3.9
Total water (L)	2.8 ± 0.79	2.4 ± 0.69	**<0.001**	2.6 ± 0.77

Each value represents the mean of up to 4 days of dietary intakes ± SD, *n* = 409. * Mann-Whitney test. Values in bold are statistically significant. PUFA, polyunsaturated fatty acids; SFA, saturated fatty acids; EPA, eicosapentaenoic acid; DHA, docosahexaenoic acid; ALA, alpha-linolenic acid; LA, linoleic acid. Values in bold are significantly different between genders.

**Table 4 ijerph-16-03036-t004:** Mean daily intake of total energy, mean percentage of energy derived from fat, and carbohydrates by quartiles of energy intake.

Intake Characteristics		First Quartile	Second Quartile	Third Quartile	Fourth Quartile	*p* for Trend *	Total
		*n* = 102	*n* = 103	*n* = 102	*n* = 102		*n* = 409
Males	Energy intake (kcal)	974.9 ± 109.49	1316.5 ± 81.93	1611.6 ± 114.21	2489.1 ± 675.23	**<0.001**	1842.1 ± 713.58
	min–max	700.3–1111.3	1137.4–1437.4	1439.9–1850.0	1860.9–6434.8		700.3–6434.8
	Percentage of energy derived from carbohydrates (%)	47.6 ± 9.34	44.2 ± 7.26	41.9 ± 7.43	41.6 ± 7.76	**0.010**	42.8 ± 7.89
	Percentage of energy derived from fat (%)	35.0 ± 7.84	36.5 ± 6.70	37.7 ± 7.42	39.3 ± 6.89	0.051	37.8 ± 7.18
Females	Energy intake (kcal)	910.0 ± 178.28	1270.5 ± 76.10	1616.8 ± 125.01	2265.7 ± 429.04	**<0.001**	1309.4 ± 463.96
	min–max	359.8–1129.7	1137.4–1439.2	1439.8–1855.5	1877.6–3770.5		359.8–3770.5
	Percentage of energy derived from carbohydrates (%)	49.4 ± 9.54	45.9 ± 8.00	44.3 ± 8.15	42.6 ± 8.07	**0.001**	46.6 ± 8.96
	Percentage of energy derived from fat (%)	33.1 ± 7.24	36.0 ± 6.04	37.1 ± 7.49	40.7 ± 7.91	**<0.001**	35.6 ± 7.40
Total	Energy intake (kcal)	922.1 ± 169.07	1289.7 ± 81.44	1614.0 ± 118.73	2440.9 ± 635.16	**<0.001**	1566.0 ± 653.42
	min–max	359.8–1129.7	1137.4–1439.2	1439.8–1855.5	1860.9–6434.8		359.8–6434.8
	Percentage of energy derived from carbohydrates (%)	49.1 ± 9.48	45.2 ± 7.70	43.0 ± 7.82	41.8 ± 7.80	**<0.001**	44.8 ± 8.66
	Percentage of energy derived from fat (%)	33.4 ± 7.35	36.2 ± 6.29	37.4 ± 7.42	39.6 ± 7.10	**<0.001**	36.7 ± 7.37

Each value represents the mean of up to 4 days of dietary intake ± SD. * 1-way ANOVA. Results in bold are statistically significant.

**Table 5 ijerph-16-03036-t005:** Prevalence of reaching the recommended intake for proportion of energy derived from fat and carbohydrates by quartiles of energy intake *.

Categories			First Quartile	Second Quartile	Third Quartile	Forth Quartile	*p*-Trend in Quartiles ***	Total
			*n* = 102	*n* = 103	*n* = 102	*n* = 102		*n* = 409
Energy derived from carbohydrates	Male	low	7 (36.8%)	25 (58.1%)	36 (65.5%)	51 (63.8%)	0.091	119 (60.4%)
		recommended	10 (52.6%)	17 (39.5%)	18 (32.7%)	28 (35.0%)		73 (37.1%)
		high	2 (10.5%)	1 (2.3%	1 (1.8%)	1 (1.3%)		5 (2.5%)
	Female	low	23 (27.7%)	27 (45.0%)	27 (57.4%)	10 (45.5%)	**0.002**	87 (41.0%)
		recommended	51 (61.4%)	31 (51.7%)	19 (40.4%)	12 (54.5%)		113 (53.3%)
		high	9 (10.8%)	2 (3.3%)	1 (2.1%)	0 (0.0%)		12 (5.7%)
	Total	low	30 (29.4%)	52 (50.5%)	63 (61.8%)	61 (59.8%)	**<0.001**	206 (50.4%)
		recommended	61 (59.8%)	48 (46.6%)	37 (36.3%)	40 (39.2%)		186 (45.5%)
		high	11 (10.8%)	3 (2.9%)	2 (2.0%)	1 (1.0%)		17 (4.2%)
		Significance level **	0.515	0.193	0.414	0.139		<0.001
Energy derived from fat	Male	low	1 (5.3%)	0 (0.0%)	0(0.0%)	0 (0.0%)	**0.074**	1 (0.5%)
		recommended	8 (42.1%)	18 (41.9%)	20 (36.4%)	19 (23.8%)		65 (33.0%)
		high	10 (52.6%)	25 (58.1%)	35 (63.6%)	61 (76.3%)		131 (66.5%)
	Female	low	3 (3.6%)	0 (0.0%)	1 (2.1%)	0 (0.0%)	**<0.001**	4 (1.9%)
		recommended	55 (66.3%)	27 (45.0%)	20 (42.6%)	5 (22.7%)		107 (50.5%)
		high	25 (30.1%)	33 (55.0%)	26 (55.3%)	17 (77.3%)		101 (47.6%)
	Total	low	4 (3.9%)	0 (0.0%)	1 (1.0%)	0 (0.0%)	**<0.001**	5 (1.2%)
		recommended	63 (61.8%)	45 (43.7%)	40 (39.2%)	24 (23.5%)		172 (42.1%)
		high	35 (34.3%)	58 (56.3%)	61 (59.8%)	78 (76.5%)		232 (56.7%)
		Significance level **	0.102	0.753	0.355	0.921		<0.001

Note * similar United States Department of Agriculture (USDA) and European Food Safety Authority (EFSA) reference range criteria: Total carbohydrates (E%): 45%–60% and Total fat (E%): 20%–35%. ** Mann-Whitney test, males versus females, per quartiles (unadjusted), and per total participants. *** Kruskal-Wallis test. Results in bold are statistically significant.

**Table 6 ijerph-16-03036-t006:** Prevalence of adult obesity reaching adequate intakes per nutrients (USDA versus EFSA) *.

Nutrient	Males		Females		Significance Level (Males vs. Females) **	
	USDA	EFSA	USDA	EFSA	USDA	EFSA
Vitamin C	27.41	18.27	22.64	13.21	0.265	0.159
Vitamin D	0.00	0.00	0.47	0.47	NA	NA
Vitamin A	14.21	22.34	14.15	15.57	0.914	0.080
Vitamin K	19.29	47.21	18.87	30.19	0.612	**<0.001**
Thiamine	74.62	97.97	45.75	99.06	**<0.001**	0.361
Riboflavin	51.27	29.95	37.74	10.85	**0.006**	**<0.001**
Niacin	61.93	90.36	38.21	86.32	**<0.001**	0.205
Vitamin B6	47.72	38.58	22.64	15.57	**<0.001**	**<0.001**
Folates	20.81	35.53	5.66	11.32	**<0.001**	**<0.001**
Vitamin B12	72.08	38.58	45.75	16.98	**<0.001**	**<0.001**
Pantothenic acid	26.40	26.40	10.38	10.38	**<0.001**	**<0.001**
Choline	2.54	14.21	2.83	2.83	0.855	**<0.001**
Vitamin E	2.03	3.05	0.00	1.89	0.053	0.448
Calcium	51.78	56.85	22.64	36.79	**<0.001**	**0.001**
Copper	57.87	12.69	33.96	8.02	**<0.001**	0.120
Fluoride	0.00	0.00	0.00	0.00	NA	NA
Iron	86.80	60.41	50.00	26.42	**<0.001**	**<0.001**
Magnesium	8.63	18.27	3.77	7.08	**0.041**	**<0.001**
Manganese	32.99	16.24	30.66	7.08	0.612	0.004
Phosphorus	83.25	93.40	54.25	77.36	**<0.001**	**<0.001**
Selenium	86.80	72.08	58.02	35.38	**<0.001**	**<0.001**
Zinc	13.20	13.20	13.21	13.21	0.998	0.998
Potassium	1.52	7.11	0.00	0.94	0.111	0.001
Sodium (bellow upper limit)	10.66	2.03	40.09	13.21	**<0.001**	<0.001
LA	3.05	6.60	0.94	5.66	0.125	0.692
ALA	2.54	4.60	1.41	5.21	0.413	0.771
DHA+EPA	14.21	14.21	8.49	8.49	0.067	0.067
Fiber	5.10	12.75	7.10	3.8	0.399	**0.001**
Protein	91.87	66.66	80.66	42.86	**0.003**	**<0.001**
Water	12.18	63.96	32.07	70.28	**<0.001**	0.173

* Each value represents the prevalence (%) of participants reaching adequate intakes by EFSA or USDA criteria, as indicated in columns. ** Chi-square test. Results in bold are statistically significant. LA, linoleic acid. ALA, alpha-linolenic acid. DHA, docosahexaenoic acid. EPA, eicosapentaenoic acid.

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
