# Peer review of "Assessment of Nutritional Intakes in Individuals with Obesity under Medical Supervision. A Cross-Sectional Study"

_ijerph, 2019, doi:10.3390/ijerph16173036_

Round 1

Reviewer 1 Report

In the manuscript "Assessment of nutritional intakes in individuals with obesity under medical supervision. A cross-sectional study", È˜erban et al. have explored the nutritional intakes in subjects with obesity in the situation of medical supervision, in the context of the Romanian NutriGen trial. This is a novel study but minor changes are needed.

Page 2, line 84, use: "other interventions such as exercise and motivational counseling or other behavioral interventions [4]."

Table 1. Define AUDIT-C.

Page 4, line 154: Use "USDA"

Page 4, Line 161 and Table 2. Use "24-h" instead of "24-hr"

Table 4. Use "Fourth".

Please, include the range (min-max) for energy intake and not solely SD.

Table 5. Define acronyms "M" and "F".

Page 10. Line 321, please rephrase.

Reviewer 2 Report

I congratulate the authors on the good work in this article. I suggest that the discussion of the results should include as a design-limit not to have dietary biomarkers, giving the opportunity for methodological validation calculations to be carried out in future analyses.

Usually when the analysis of sodium consumption is performed in healthy populations, underestimation is present, which will have to be discussed considering the non-interpretation of biomarkers. 

Finally, dietary assessment in obese subjects (even more under an attempted hypocaloric diet) may suggest sub-under-reporting in the consumption of certain food groups, usually processed or with high energy denisity. Dietary analysis could be enriched by presenting the list of foods grouped according to its main component.  
